# First-Principles Study on the Effect of Lithiation in Spinel Li_x_Mn_2_O_4_ (0 ≤ x ≤ 1) Structure: Calibration of CASTEP and ONETEP Simulation Codes

**DOI:** 10.3390/ma15165678

**Published:** 2022-08-18

**Authors:** Donald Hlungwani, Raesibe Sylvia Ledwaba, Phuti Esrom Ngoepe

**Affiliations:** Materials Modelling Centre, University of Limpopo, Private Bag X1106, Sovenga 0727, South Africa

**Keywords:** LiMn_2_O_4_, density functional theory, energy storage, linear-scaling DFT, lithium intercalation, lithium-ion battery

## Abstract

Lithium–manganese–oxide (Li-Mn-O) spinel is among the promising and economically viable, high-energy density cathode materials for enhancing the performance of lithium-ion batteries. However, its commercialization is hindered by its poor cyclic performance. In computational modelling, pivotal in-depth understanding of material behaviour and properties is sizably propelled by advancements in computational methods. Hence, the current work compares traditional DFT (CASTEP) and linear-scaling DFT (ONETEP) in a LiMn_2_O_4_ electronic property study to pave way for large-scale DFT calculations in a quest to improve its electrochemical properties. The metallic behaviour of Li_x_Mn_2_O_4_ (0.25 ≤ x ≤ 1) and Li_2_Mn_2_O_4_ was correctly determined by both CASTEP and ONETEP code in line with experiments. Furthermore, OCV during the discharge cycle deduced by both codes is in good accordance and is between 5 V and 2.5 V in the composition range of 0 ≤ x ≤ 1. Moreover, the scaling of the ONETEP code was performed at South Africa’s CHPC to provide guidelines on more productive large-scale ONETEP runs. Substantial total computing time can be saved by systematically adding the number of processors with the growing structure size. The study also substantiates that true linear scaling of the ONETEP code is achieved by a systematic truncation of the density kernel.

## 1. Introduction

Deepening the understanding of multiscale physical modelling techniques, spanning from the electronic to atomistic modelling level, can explicate key fundamentals about the materials’ behaviour, aging and subsequently give insights on how optimal specifications affect the design and how such designs govern the operation of devices [1]. A detailed understanding of lithium-ion battery’s operational mechanisms is essential for accelerating the improvement in and development of highly efficient energy storage systems for large-scale applications, particularly automotive and smart grids [2]. The physical theory and computational electrochemistry provide an in-depth understanding of mechanisms occurring at multiple spatio-temporal scales in a single material or component or combinations thereof [1]. The development of new sophisticated computational methods and an increase in computational power plays an important role in the success of computational modelling of materials.

Quantum chemical models (first principles or ab initio) techniques based on electronic theories, that do not rely on any parameters, play a significant role by suggesting guidelines to improve well-known active materials and subsequent discovery of new ones, with specific functionality by exploiting the fundamental laws governing their behaviour [1,3,4,5]. Aydinol M. K, Kohan A. F and Ceder G used first-principles studies to determine lithium intercalation voltages of metal oxides which are prospective electrodes for the emerging high-energy lithium-ion batteries. The intercalation voltage is one of the factors that affect the total energy density of a battery; hence, this was one of the monumental findings of first-principles computational studies [6].

Moreover, density functional theory was also employed to study structural and electronic properties of binary and ternary spinel oxides by Walsh A and co-workers, wherein, bandgap analysis with respect to Al substitution into the spinel systems was carried out to tune electronic properties of materials [7]. For more accurate results, many co-existing methods are being tuned in order to attain more sophisticated computational methods for a far greater impact in science. This shows some of the work done with several quantum mechanics methods evincing the impact and success of DFT. However, most traditional DFT codes that are based on delocalized pseudopotential wave functions are computationally expensive, they scale cubically with system size. This constrains the number of atoms that can be handled by the current computer systems to a few hundred. The exciting properties of nanostructured materials constituting thousands of atoms that affect lithium-ion battery performance calls for a description at a quantum mechanics level. This will accelerate the improvement of lithium-ion battery performance, and linear scaling DFT methods can play a pivotal role in this regard.

Substantial research efforts have been delegated in fine-tuning some of the current methods implemented in various traditional DFT codes to achieve linear scaling [8,9,10]. Considerable computational time can be saved by making use of localized electronic basis sets on systems characterized by a fixed bandgap. The Order-N Electronic Total Energy Package (ONETEP) is one of the linear-scaling DFT packages that are taking full advantage of high-performance parallel computers. This is achieved by truncating the density matrix to a highly localized density matrix expressed by nonorthogonal, highly localized basis functions.

The spinel LiMn_2_O_4_ is presently the centre of much interest as a cathode material for high-power lithium batteries due to the direct impact that the cathode material has on the safety and capacity of LIBs [11,12]. The low manufacturing cost, low toxicity, and high charge/discharge rate capability of spinel LiMn_2_O_4_ (LMO) have rendered it advantageous amongst other oxide electrodes [13,14]. However, their commercialization has been long delayed mainly due to disproportionation reaction and phase transformations resulting in capacity fading [15,16,17]. The disproportionation reaction results in Mn^2+^ and Mn^4+^ according to the reaction: 2Mn^3+^ → Mn^2+^ + Mn^4+^. Mn^2+^ is soluble in acidic electrolytes leading to capacity fading during cycling. A cubic to tetragonal phase transition has been noted in the literature, associated with more than 50% concentration of Mn^3+^ (t_2g_^3^_g_^1^) in LiMn_2_O_4_ which exacerbate manganese dissolution. Xu Li et al. employed first-principles studies to explore defects in LiMn_2_O_4_ spinel by monitoring their charge compensation mechanism. The study revealed the presence of Li, O, and Mn vacancy defects, and Li and Mn interstitial point defects [18]. The essential contribution of first-principles studies to the research of energy storage systems has also been demonstrated in a study by Ma Y and co-workers in investigating the effects of doping LiMn_2_O_4_ spinel with Ti. The Ti dopant in LiMn_2_O_4_ is reported to be in a 4+ valence state, and thus stabilizes the spinel structure [19]. Such quantum mechanics-based explorations are pivotal to energy storage systems and can extend their pivotal role in enhancing the performance of spinel LiMn_2_O_4_. However, these studies were only carried out for a limited number of atoms (about a few hundred atoms) due to inherent cubic scaling in traditional DFT codes. Studies of molecular dynamics simulated Li-Mn-O spinel nanostructures consisting of thousands of atoms with exciting myriads structural features are not possible with traditional DFT. Moreover, elucidation of the effect of intrinsic defects contained in these Li-Mn-O nanoarchitectures such as grain boundaries, vacancy, and interstitial structural defects is essential for enhancing their performance in lithium-ion batteries [20,21,22]. The ONETEP linear-scaling DFT code is one of the codes that are capable of describing the interaction of thousands of atoms at the quantum mechanics level. However, the accuracy of a true linear scaling calculation relies on many parameters which require tedious fine-tuning.

Hence, in this work, we perform DFT calculations using the CASTEP (traditional DFT) and the ONETEP (linear-scaling DFT) codes elucidating the effect of lithiation on the electronic structure of spinel LiMn_2_O_4_ and the scaling of the ONETEP code on the supercomputers of the Centre for High Performance Computing (CHPC) to determine optimal parameters required to study the Li-Mn-O nanoarchitectures comprising of thousands of atoms with the ONETEP code. The study will serve as a guide in moving from traditional DFT into linear-scaling DFT and on how to adequately allocate computing resources when using the ONETEP code.

## 2. Materials and Methods

First-Principles Electronic Structure Calculations: The density functional theory (DFT) calculations have been performed using the CASTEP [23] and the ONETEP [24] simulation codes with a plane-wave energy cutoff of 800 eV. The generalized gradient approximation (GGA) of Perdew−Burke−Ernzerhof (PBE) [25] was selected as the exchange-correlation functional. The ultrasoft pseudopotentials [26] were employed for the CASTEP code and the projector augmented wave pseudopotentials [27] were used for the ONETEP code_._ The cell, atomic positions and the lattice parameters of all the Li-Mn-O spinel conventional unit cells were fully optimized with the CASTEP code. The geometry optimization convergence energy tolerance was set to 2.0 × 10^−5^ eV/atom and the Self-consistent field (SCF) for controlling the electronic minimization algorithm was set to 1.0 × 10^−6^. Moreover, the Broyden-Fletcher-Goldfarb-Shanno (BFGS) algorithm with a force and stress convergence tolerance of 0.05 eV/Å and 0.1 GPa was used to ensure successful optimization. Furthermore, a 4 × 4 × 4 Monkhorst pack k-mesh sampling was used to sample the Brillouin zone with a width of 0.05 eV. For the comparative calculations of CASTEP and ONETEP, the Li-Mn-O spinel structures is initially optimized using the CASTEP code before undergoing the discharge process. The NGWF radii of 7.5 Å was used for all the ONETEP calculations. The same convergence tolerance of 2.0 × 10^−5^ eV/atom and electronic energy minimization tolerance of 1.0 × 10^−6^ eV/atom used on the CASTEP code were also adopted for the ONETEP calculations.

ONETEP Scaling Tests: The propitious LiMn_2_O_4_ lithium-ion battery cathode material was used to perform the ONETEP scaling tests. Different system configurations of the spinel LiMn_2_O_4_ structure were chosen in accordance with the type of scaling. The system size was varied from 112–1512 atoms. The calculations are performed on the Lengau cluster of the Centre for High Performance Computing (CHPC) based in Cape Town. The cluster consists of 1368 nodes connected to shared storage of 4 Petabytes through FDR InfiniBand Network interconnect. Each compute node houses 24 processors of a 2.6 GHz CPU clock. Moreover, the Lengau cluster is scored a LINPACK Benchmark Rpeak value of 1.307 and Rmax value of 1.029 PFlops.

Structural Properties: The conventional unit cell of spinel LiMn_2_O_4_ comprise of 8 lithium (Li) atoms, 16 manganese (Mn) atoms and 32 oxygen (O) atoms wherein the Li atoms occupy the 8a tetrahedral sites, Mn atoms situated at the 16d octahedral sites and O atoms are situated at the 32e sites. Upon full lithiation of Li_x_Mn_2_O_4_, lithium ions start occupying 16c octahedral sites as illustrated in Figure 1. The MnO_6_ framework of spinel forms a close-pack cubic array [28,29], resulting in three-dimensional lithium diffusion channels. The LiMn_2_O_4_ spinel structure crystallizes into a cubic structure with a space group of Fd-3m at room temperature.

## 3. Results

### 3.1. Structural Properties and Electronic Properties

Table 1 depicts the lattice constants for spinel structures after geometry optimization in the current work. The lattice values are compared to theoretical values from Quantum Expresso [30] and experimental values [31]. The lattice values depict good comparison from experimental work, i.e., 8.297 Å with a difference of 0.59% from the experimental value and a difference of 1.66% from GGA was obtained for LiMn_2_O_4_. Similarly, the Li_2_Mn_2_O_4_ shows reasonable agreement of lattice values of 8.231 Å compared to the lattice parameters from literature calculated using VASP code [32], with a difference of 1.79% from GGA for Li_2_Mn_2_O_4_. Figure 2 shows the MnO_6_ octahedron of the optimized LiMn_2_O_4_ spinel structure and the Mn–O bond length and the O–Mn–O angles of the Mn^4+^ and Mn^3+^ states are shown. The Mn^4+^–O_6_ set of bond distances do not veer much from the Mn^3+^–O_6_ set of bond distances (~1.8665 Å) indicating the absence of the Jahn–Teller distortion in the 0 ≤ x ≤ 1 lithium concentration region. Moreover, the electronic configuration of Mn^3+^ is t_2g_^3^e_g_^1^ and of Mn^4+^ is t_2g_^3^e_g_^0^ are shown in the figure. The O–Mn^4+^–O angle is less than the O–Mn^3+^–O angle owing to the presence of the O–Li bond around the Mn^3+^O_6_ octahedron.

Density of States (DoS): Figure 3i shows the total DOS with contribution from spin-up and spin-down states of Figure 3(a) LiMn_2_O_4_ from literature, Figure 3(b) LiMn_2_O_4_ (this work), and Figure 3(c) Li_2_Mn_2_O_4_ (this work). The density of states in Figure 3(b) are in good agreement with the density of states depicted in Figure 3(a) which were calculated by S. Baǧcı and co-workers [30]. The Fermi level of all the presented structures in Figure 3i lies on the sharply increasing peak of the e_g_ spin-up states illustrating metallic behaviour, which is in great accordance with the findings of Xu et al. [33]. The Fermi level is dominated by the spin-down states in both LiMn_2_O_4_ and Li_2_Mn_2_O_4_. The number of occupied states increases with an increase in lithium content, evidenced by the broadening of t_2g_ orbitals with spin-down states below the Fermi level. Moreover, the Fermi level which cuts the peak denoting the t_2g_ orbitals with spin-down states at ~−18.18 for LiMn_2_O_4_ and ~−27.79 for Li_2_Mn_2_O_4_ indicates an increase in lithium content. The e_g_ states comprising of spin-down electronic states are empty, indicating that for this material in this orbital, electrons are in high spin-up states.

The partial density of states depicted in Figure 3i represents contributions of the Figure 3(a) Lithium, Figure 3(b) Oxygen, Figure 3(c) Manganese atoms in Figure 3(1) LiMn_2_O_4_ and Figure 3(2) Li_2_Mn_2_O_4,_ respectively. The total density of states for both systems, depict the metallic behaviour of LiMn_2_O_4_ and Li_2_Mn_2_O_4_, which emanates from the Mn 3d orbitals with minimal contribution from the O 2p orbitals. An intense non-bonding peak located approximately between −18 eV and 15 eV originating from the O 2s orbitals is noted on the density of states of both LiMn_2_O_4_ and Li_2_Mn_2_O_4_. The Mn 3d density of states of Li_2_Mn_2_O_4_ on the pseudogap formed between the t_2g_ and e_g_ orbitals just below the Fermi level are higher than that of LiMn_2_O_4_ indicating an increase in the number of allowed states in the pseudogap with lithiation. The energy range between −8 eV to 2.8 eV is dominated by the O 2p and Mn 3d orbitals indicating a strong hybridization between manganese and oxygen atoms in both LiMn_2_O_4_ and Li_2_Mn_2_O_4_**.**

### 3.2. Discharging the Li_x_Mn_2_O_4_ (0 ≤ x ≤ 1) Spinel

The structure of spinel Li_x_Mn_2_O_4_ at different lithium concentrations indicating the discharging process of this electrode material is shown in Figure 4a. Figure 4(ai) Li_0.25_Mn_2_O_4_, Figure 4(aii) Li_0.5_Mn_2_O_4_, Figure 4(aiii) Li_0.75_Mn_2_O_4_, and Figure 4(aiv) Li_1.0_Mn_2_O_4_ structures were captured to represent the lithium concentrations between 0 and 1 in the Li-Mn-O spinel structures, in which the concentration of the Jahn–Teller active Mn^3+^ ions is minimal. The intercalation of lithium atoms into the 8a tetrahedral sites reduces the number of Mn^4+^ in the system as a lithium atom donates an electron to an oxygen atom. The number of Mn^4+^ ions in the Li_1.0_Mn_2_O_4_ structure is equal to the number of Mn^3+^ ions; as such, the further addition of lithium atoms from this point will facilitate the dominance of the Jahn–Teller active Mn^3+^ ions. Figure 4b shows the oxygen tetrahedron Figure 4(bi) before and Figure 4(bii) after the addition of a lithium atom. This illustrates the mechanism of the discharge process at a tetrahedral site in the lithium–manganese–oxide spinel structure. Before lithiation, the distance between oxygen atoms in the oxygen tetrahedron is approximately 3.3 Å and after lithiation, the distance is approximately 3.2 Å indicating the overlap of the lithium S and oxygen P orbitals evincing the existence of Li-O bond and hence a successful lithiation.

In Figure 5, we illustrate the electrostatic potential in spinel Li_x_Mn_2_O_4_ (0 ≤ x ≤ 1) structure given by the Kohn-sham potential essential for indicating probably sites for lithium intercalation in the lithium composition range between 0 and 1. The electrostatic potential field in the figure shows the potential energy of a positive charge in the vacant sites of the Li_x_Mn_2_O_4_ (0 ≤ x ≤ 1) structure. The isosurface shows the electrostatic potential at 8a tetrahedral sites and 16d octahedral sites in this spinel structure. The potential field is more pronounced in the 8a tetrahedral sites than in the 16d octahedral sites substantiating the observed 16d position of lithium atoms in spinel Li_x_Mn_2_O_4_ (0 ≤ x ≤ 1) [34,35]. The lithiation into the 8a tetrahedral sites is shown in Figure 5a–e. Figure 6 shows the potential field of the topotactical delithiated lithium–manganese–oxide spinel, which is at the highest in the lithium composition range (0 ≤ x ≤ 1), this is indicated by the strong electrostatic potential fields illustrated in the structure. Moreover, Figure 5a shows the Li composition of this spinel structure in which all the 8a tetrahedral sites are occupied and the potential field is only observed at the 16c octahedral sites.

### 3.3. CASTEP and ONETEP Open Cell Voltage (OCV) of the Li_x_Mn_2_O_4_ (0 ≤ x ≤ 1) Spinel

Figure 6 illustrates the Open Cell Voltage (OCV) computed using CASTEP and ONETEP simulation codes of the spinel Li–Mn–O structure in the lithium concentration range between 0.25 and 1. The OCV is calculated using the following equation:(1)$$Voltage=-\frac{\mu_{Li}^{cathode}-\mu_{Li}^{anode}}{ze}$$

In which $\mu_{Li}^{cathode}$ is the chemical potential of lithium atoms in the cathode, $\mu_{Li}^{anode}$ is the chemical potential of lithium atoms in the anode, e   is the charge of an electron and z represent the valence of the ion. For a system under constant pressure and temperature, the Li chemical potential in the cathode can be given by the change in Gibbs free energy, therefore:(2)$$Voltage=-\frac{\mu_{Li}^{cathode}-\mu_{Li}^{anode}}{ze}=-\frac{\frac{dG}{dN_{Li}}-\mu_{Li}^{anode}}{e}$$
where G is the Gibbs free energy.

The CASTEP and ONETEP calculated voltages in the lithium composition range between 0.25 and 1 are ~5.1 V and ~4.4 V, respectively. The intercalation voltages are averaged by Li_0.25_Mn_2_O_4_, Li_0.5_Mn_2_O_4_, Li_0.75_Mn_2_O_4_ and Li_1.0_Mn_2_O_4_; hence, only one plateau is observed in this region, which is between 0.5 and 0.75 lithium composition range. Moreover, Li_x_Mn_2_O_4_ (0 ≤ x ≤ 1) maintained its cubic structure as such the latter mentioned plateau is due to the reported topotactical phase change [36,37]. For 0.75 < x < 1 in Li_x_Mn_2_O_4_, the ONETEP calculated voltage then drops significantly to ~2.6 V whilst the CASTEP calculated voltage drops to ~3.2 V. Lithium–lithium repulsion increases with the filling of the tetrahedral sites causing lithium ordering. This step in this region (0.75–1.00) is attributed to Li ordering [17,38]. The ONETEP calculated OCVs are more comparable to OCVs calculated by F.F Bazito and R.M Torresi [39].

**Figure 6 materials-15-05678-f006:**
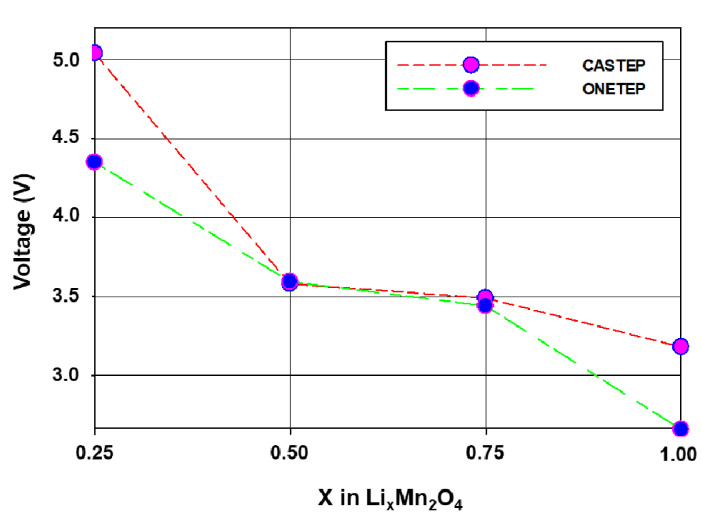
Comparison of ONETEP and CASTEP Open Cell Voltage (OCV) Li_x_Mn_2_O_4_ (0.25 ≤ x ≤ 1).

DoS of Li_x_Mn_2_O_4_ (0 ≤ x ≤ 1) calculated with CASTEP and ONETEP: The electronic structure of spinel Li_x_Mn_2_O_4_ with a lithium concentration range of 0 to 1 is illustrated by the DoS in Figure 7 and Figure 8. Figure 7 shows the DoS of Mn_2_O_4_, Li_0.5_Mn_2_O_4_, and Li_1.0_Mn_2_O_4_ calculated using the CASTEP code. The DoS of the topotactically delithiated spinel Mn_2_O_4_ in Figure 7a display a bandgap of ~0.65 eV. Li_0.5_Mn_2_O_4_ and Li_1.0_Mn_2_O_4_ exhibit metallic behaviour indicated by the overlapping valence and the conduction band in the DoS shown in Figure 7b,c. Figure 7d shows the effect of lithiation on the electronic structure of the Li-Mn-O spinel. Most of the Li_1.0_Mn_2_O_4_ states are below zero energy than most of the Li_0.5_Mn_2_O_4_ and Mn_2_O_4_ states.

The linear scaling ONETEP code was also used to scrutinize the electronic structure of the Li-Mn-O spinel during lithium intercalation as captured by DoS in Figure 8. Particularly, the significant change in the charge state of manganese in the structure during lithiation. The electronic configuration of Mn^3+^ ions is t_2g_^3^e_g_^1^ which increases with lithium intercalation and for Mn^4+^ is t_2g_^3^e_g_^0^ which decreases with lithium intercalation. For Figure 8a Mn_2_O_4_, all the t_2g_ spin-up states are occupied, followed by a bandgap of ~0.58 eV. However, with the addition of lithium atoms, the e_g_ spin-up states and the t_2g_ spin-down states are occupied as shown by the DoS of Figure 8b Li_0.5_Mn_2_O_4_, and Figure 8c Li_1.0_Mn_2_O_4_. As such, the valence band and the conduction band overlap increases with increasing lithium content.

### 3.4. Scaling of the ONETEP Code South Africa’s CHPC

Computational work that can be efficiently divided among several processors facilitated by sophisticated message-passing standards can be performed swiftly in high-performance computing systems. In such a case, increasing the number of processors results in increased computational power and a considerable computing time can be saved. The scalability of a computational code depends significantly on the parallel algorithms implemented in the code. Hence, in poorly parallel computational codes increasing the number of processors does not reduce the computational time. In this section, we evaluate the performance of the ONETEP code by performing scaling tests to gain insights into the allocation of the precious computational resources at South Africa’s CHPC. Strong and weak scaling tests will be performed, in a strong scaling test, the problem size is fixed and the number of processors is increased. For a weak scaling test, the problem size and the number of processors are both systematically increased. The speedup is simply calculated as the ratio of the time it takes to run on a single processor and the time it takes to run on N processors as indicated by Equation (3) below.
(3)$$\mathrm{Speedup}=\frac{T\left(1\right)}{T\left(N\right)}$$
where T(1) is the runtime on a single processor and T(N) is the runtime on N processors.

Figure 9 illustrates the strong scaling of the ONETEP code on the Centre for high performance computing (CHPC) cluster performed with the spinel LiMn_2_O_4_ (112, 448, 1512) as the input structures. The number of processors is varied by 24 because each compute node consists of 24 processors on the cluster. In the figure, the three speed-up curves of LiMn_2_O_4_ are compared to an ideal speed-up curve. The calculated speed-up values for LiMn_2_O_4_ (112, 448, and 1512 atoms) in the figure are comparable for the number of processors that are between 1 and 48. The curves in the 1 to 48 number of processors range are close to the ideal speed-up curve, and they start to veer from this curve at 48 number of processors. A significant difference in the speed-up values of the three LiMn_2_O_4_ curves is noted from 48 processors. The 112 atoms structure has the lowest speed-up values which are increasing infinitesimally with computing resources between 48 and 96 processors. This shows that the structure is small for the number of processors that are greater than 48. A constant significant increase in the speed-up of the 448 atoms structure is observed in the latter mentioned number of processor range. The 1512 atoms structure has high speed-up values in the range between 72 and 96 processors indicating that the structure was effectively shared amongst the processor than the 112 and 448 structures in this range of processors. In Figure 10, the problem size and the number of processors were both varied to give insights on the allocation of computing resources concerning system size. The kinetic energy of the psinc basis sets was set to 800 eV, and the NGWF radii to 7.5 Å. The total time is increasing linearly as a function of the number of atoms and processors as evinced by the straight-line graph depicted in the figure. This suggests that practical computing time can be achieved by systematically increasing the number of processors as the problem size grows.

A substantial amount of computational time can be saved by carefully reducing the value of the kernel cut-off, which offers a way to truncate the density matrix. Figure 11 shows a graph of time as a function of kernel cut-off for a 1296 atoms Mn_2_O_4_ spinel structure. Single point energy calculations were performed using a 7.5 Å NGWF radius and 800 eV cut-off energy. The kernel cut-offs were set to 40, 60, 80, 200, 400, 600, 800, and 1000 Bohr. The calculations with kernel cut-off between 40 and 200 Bohr took approximately 4.6 h and calculations with kernel cut-off between 400 and 1000 Bohr took approximately 7 h. The accuracy of the results was preserved as shown on the total energy vs. kernel cut-off and NGWF RMS gradient vs. kernel cut-off plots in Figure 12. In which, the total energy and the NGWF RMS gradient are the same for all the kernel cut-offs (40–1000 Bohr). This provides an opportunity to save computational time whilst preserving the accuracy of the results. The difference between the computational time plateaus is 2.5 h, which is what can be saved in this setup by using a kernel cut-off between 40 and 200 Bohr.

## 4. Discussion

In the current study, we have captured the electronic structure of Li_x_Mn_2_O_4_ spinel through Density of States (DoS). The spinel Li_x_Mn_2_O_4_ structure was relaxed with the CASTEP code and it yielded lattice constants with reasonably comparable values with experiments within a range of 2% for both systems. The DoS in Figure 3(a) depicted metallic behaviour for both LiMn_2_O_4_ and Li_2_Mn_2_O_4_. This behaviour is validated by findings from previous work by S. Baǧcı and co-workers [30]. Moreover, the filling of the t_2g_ orbitals with spin-down states increases with lithiation, and the spin-up states of the t_2g_ orbitals are filled. Moreover, the Fermi energy cuts the spin-up DoS curve on the sharply increasing curve depicting the e_g_ spin-up orbitals. This indicates the presence of the Mn^3+^ with an electronic configuration of (t_2g_ [↑↑↓]-e_g_ [↑]). Figure 3(b) indicates that the metallic behaviour of Li_x_Mn_2_O_4_ (1 ≤ x ≤ 2) spinel is a consequence of manganese and oxygen atoms. Studies of the intercalation of lithium atoms into spinel Li_x_Mn_2_O_4_ (0 ≤ x ≤ 1), which is reported to contain between 0 and 50% of the Mn^3+^ ions associated with the Jahn–Teller distortion and disproportionation reaction. In this range (0 ≤ x ≤ 1), lithium atoms occupy the 8a tetrahedral sites this is in line with the electrostatic potential shown in Figure 5. Figure 5a depicts some of the crucial stages of the discharge process in the range between 0 and 1 and Figure 5b indicates the Li-O bond evinced by a shorter distance between oxygen atoms in oxygen tetrahedron with a lithium atom in the tetrahedral position than the distance between oxygen atoms in such tetrahedron with a vacant tetrahedral site. This demonstrates successful lithium intercalation in the spinel structure and hence the discharge process.

In a quest for a swift adaptation of linear-scaling DFT techniques, particularly ONETEP for more practical and accurate results, in this section we discuss the electronic properties of Li_x_Mn_2_O_4_ (0 ≤ x ≤ 1) calculated with both CASTEP and ONETEP. In Figure 6, CASTEP and ONETEP calculated discharge OCVs of Li_x_Mn_2_O_4_ in the composition range 0 and 1 have been compared. The Li_x_Mn_2_O_4_ curve was averaged by the four points, 0.25, 0.5, 0.75, and 1.0, representing lithium composition in the Li-Mn-O spinel structure. The two curves indicate a plateau at 0.5 and 0.75 which corresponds to the reported 4 V plateau in Li_x_Mn_2_O_4_ compounds. The general shape of CASTEP and ONETEP OCV discharge curves in this work compares well with calculated Li_x_Mn_2_O_4_ discharge curves in the literature [39,40]. Moreover, the electronic structure of Li_x_Mn_2_O_4_ in this composition region (0 ≤ x ≤ 1) was also calculated with both codes (CASTEP and ONETEP). In Figure 7, the effect of adding lithium atoms in the Li_x_Mn_2_O_4_ (0 ≤ x ≤ 1) structure is successfully captured by the CASTEP code. The ~0.58 eV bandgap of Mn_2_O_4_ which is in line with experiments was deduced and the metallic nature of LiMn_2_O_4_ is also in line with literature. The same results were reproduced by the ONETEP code as depicted by the DoS in Figure 8. Moreover, the filling of the t_2g_ spin-down states with lithium intercalation correspond to the electronic configuration of (↑↑↓) for the t_2g_ orbitals from (↑↑↑), evincing a decrease in the magnetic moment of the material.

The scaling of the ONETEP code at CHPC’s Lengau cluster was carried out successfully with a number of processors ranging from 1 to 96 for the 112, 448, and 1512 atoms spinel LiMn_2_O_4_ supercells. The increase in speed-up with the number of processors for the 112 atoms LiMn_2_O_4_ supercell drops from 48 processors, indicating that from this point adding more resources will not result in less computing time. A significant amount of computational time is spent on communications between processors. However, for the 1512 atoms LiMn_2_O_4_ supercell, the speed-up values increase almost linearly with the number of processors. Total energy calculations for spinel Mn_2_O_4_ superstructures ranging from 96 to 3072 atoms with processors varying from 24 to 216 cores have been performed and the corresponding time and number of atoms plot is given in Figure 10. The number of processors was systematically varied from 24 to 216 with system size (96 to 3072), the total time increased linearly with the increasing number of atoms and number of processors. This suggests that practical computational time can be achieved by systematically increasing computing resources as the problem size grows. A substantial amount of computational time can be saved by using a cut-off of less than 400 Bohr for the Mn_2_O_4_ structures whilst preserving the accuracy of the results, as indicated in Figure 11 and Figure 12.

## 5. Conclusions

In the current study, a detailed theoretical analysis of structural and electronic properties of spinel LiMn_2_O_4_ utilizing CASTEP and ONETEP codes is presented. Electronic structure changes were captured by DoS, in which the metallic nature of LiMn_2_O_4_ was deduced with both simulation codes. The filling of the valence band with lithium intercalation into the spinel Li_x_Mn_2_O_4_ (0 ≤ x ≤ 1) structure was observed. Moreover, discharging open-circuit voltages (OCVs) were also calculated, and found to compare well with experiments. The ONETEP and CASTEP calculated results are in a great accord. Our results show the absence of the Jahn–Teller distortion in the 0 ≤ x ≤ 1 lithium concentration region and as such, LiMn_2_O_4_ can offer durable performance when cycled in this region. The ONETEP scaling tests performed in this study indicate that a considerable amount of computing time could be saved by carefully assigning computing resources, and DFT studies of large-scale systems are possible with the ONETEP code. Our current findings form a basis for moving from traditional DFT to linear-scaling DFT which will enable future studies of the effect of intrinsic structural defects generated by the amorphization and recrystallization technique on the operating voltage and the electronic structure of different Li-Mn-O nanospherical spinel structures consisting of thousands of atoms. These nanostructures contain essential structural features which are comparable to structural features observed in real materials (materials synthesized in traditional experiments); hence, their results are of great importance.

## Figures and Tables

**Figure 1 materials-15-05678-f001:**
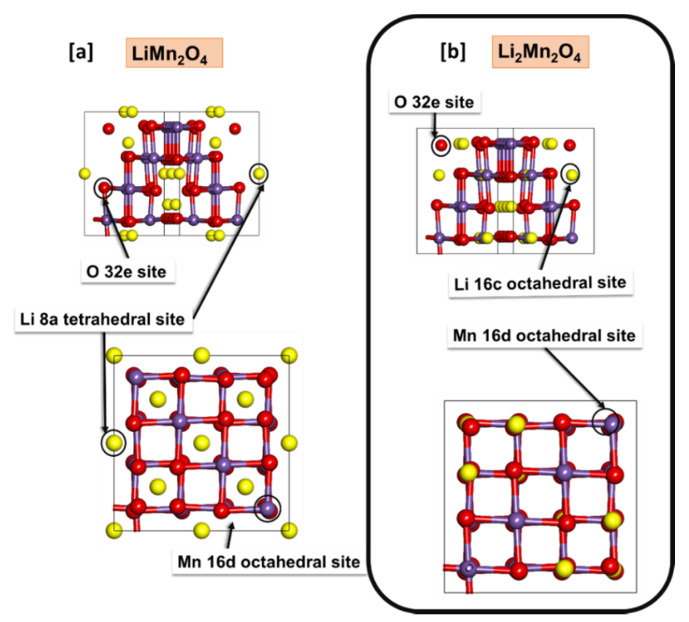
Conventional unit cells of (**a**) cubic LiMn_2_O_4_ and (**b**) cubic Li_2_Mn_2_O_4_ depicting the Wyckoff positions of lithium (yellow), manganese (purple) and oxygen (red) atoms in 8a or 16c, 16d and 32e sites, respectively.

**Figure 2 materials-15-05678-f002:**
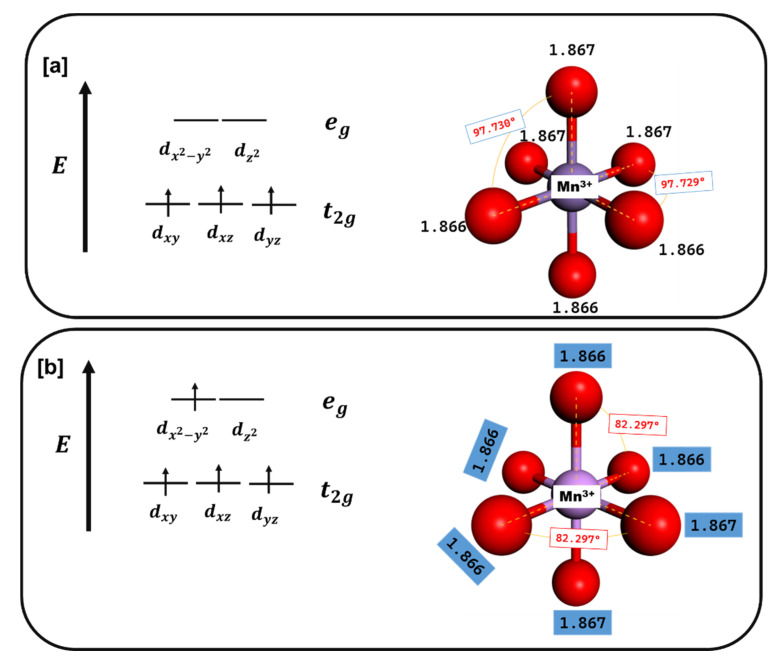
The manganese and oxygen octahedron in the LiMn_2_O_4_ spinel framework showing the (**a**) Mn^4+^ and (**b**) Mn^3+^ states. Illustration of the Mn–O bond distances, the O–Mn–O angles capturing the MnO_6_ octahedral environment and the electronic configuration of Mn^4+^ and Mn^3+^.

**Figure 3 materials-15-05678-f003:**
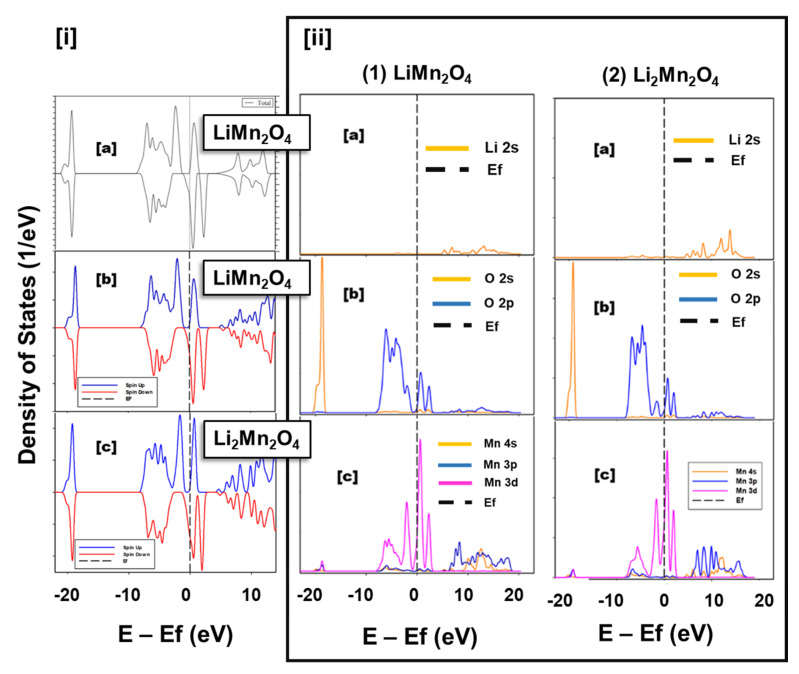
(**i**) Illustrates Spin polarized total density of states (DoS) for cubic spinel (a) LiMn_2_O_4_ (literature) [30], (b) LiMn_2_O_4_ (this work) and (c) Li_2_Mn_2_O_4_ (this work); (**ii**) Partial DoS for spin polarised cubic spinel (1) LiMn_2_O_4_ and (2) Li_2_Mn_2_O_4_, with (a) contributions from lithium, (b) oxygen and (c) manganese atoms. Figure 3(ia) reprinted with permission from Ref. [30]. 2022, Elsevier.

**Figure 4 materials-15-05678-f004:**
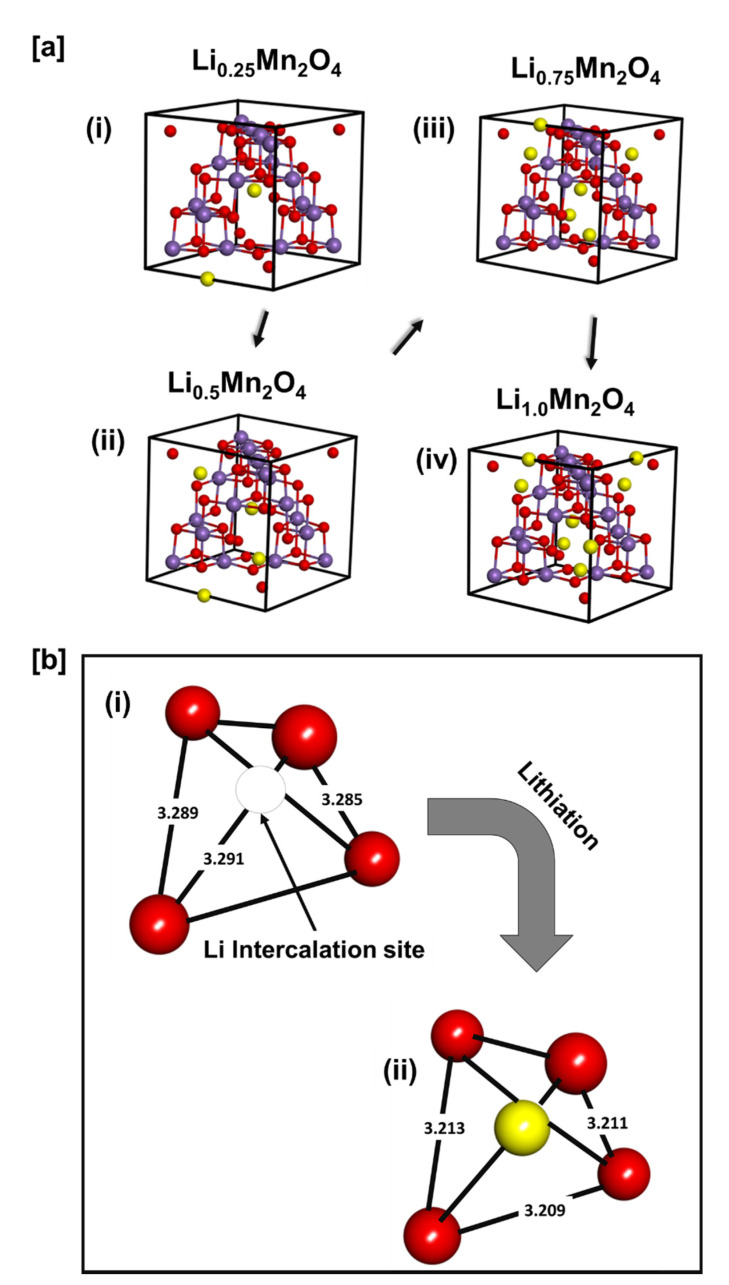
(**a**) The Intercalation of lithium atoms into the tetrahedral sites of spinel Li_x_Mn_2_O_4_ (0.25 ≤ x ≤ 1). (i) Li_0.25_Mn_2_O_4_, (ii) Li_0.5_Mn_2_O_4_, (iii) L_i0.75_Mn_2_O_4_ and (iv) Li_1.0_Mn_2_O_4_; (**b**) Lithium intercalation into a (i) vacant tetrahedral site of spinel Li_x_Mn_2_O_4_ (0 ≤ x ≤ 1) structure.

**Figure 5 materials-15-05678-f005:**
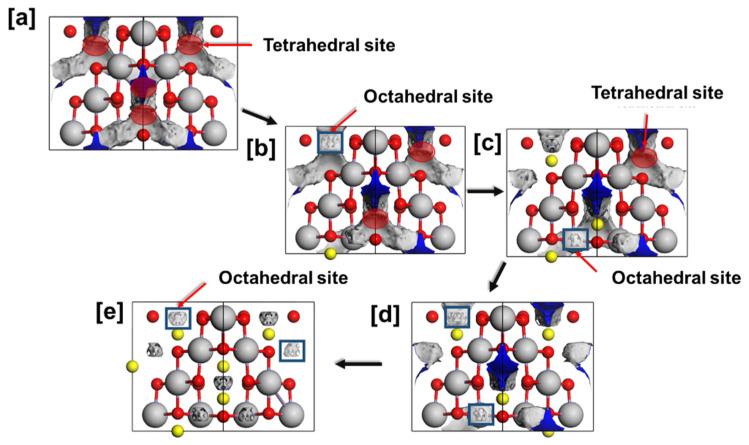
The variation in the electrostatic potential at different lithium concentrations in the spinel Li_x_Mn_2_O_4_ (0 ≤ x ≤ 1) structure. (**a**) λ-Mn_2_O_4_, (**b**) Li_0.25_Mn_2_O_4_, (**c**) Li_0.5_Mn_2_O_4_, (**d**) L_i0.75_Mn_2_O_4_ and (**e**) Li_1.0_Mn_2_O_4_.

**Figure 7 materials-15-05678-f007:**
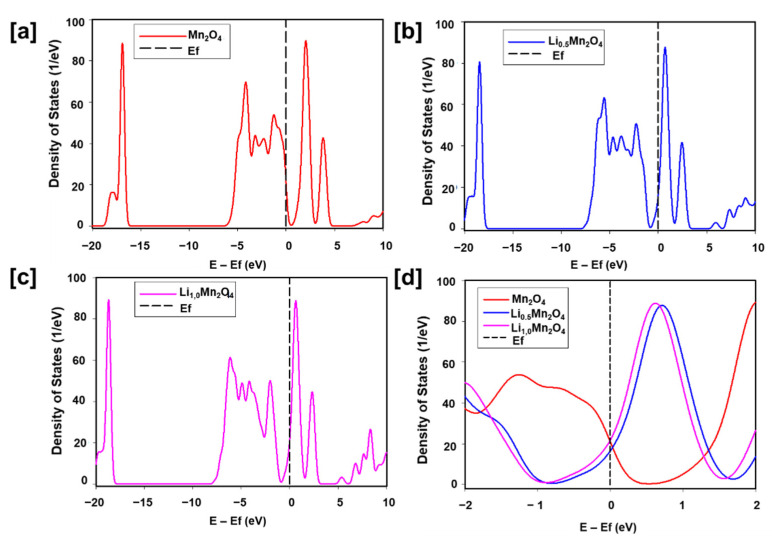
Changes in DoS of Li-Mn-O spinel during lithium intercalation for (**a**) Mn_2_O_4_, (**b**) Li_0.5_Mn_2_O_4_ and (**c**) Li_1.0_Mn_2_O_4_ with CASTEP (Cambridge Serial Total Energy Package). (**d**) Comparison of DoS of Mn_2_O_4_, Li_0.5_Mn_2_O_4_ and Li_1.0_Mn_2_O_4_ at the fermi energy.

**Figure 8 materials-15-05678-f008:**
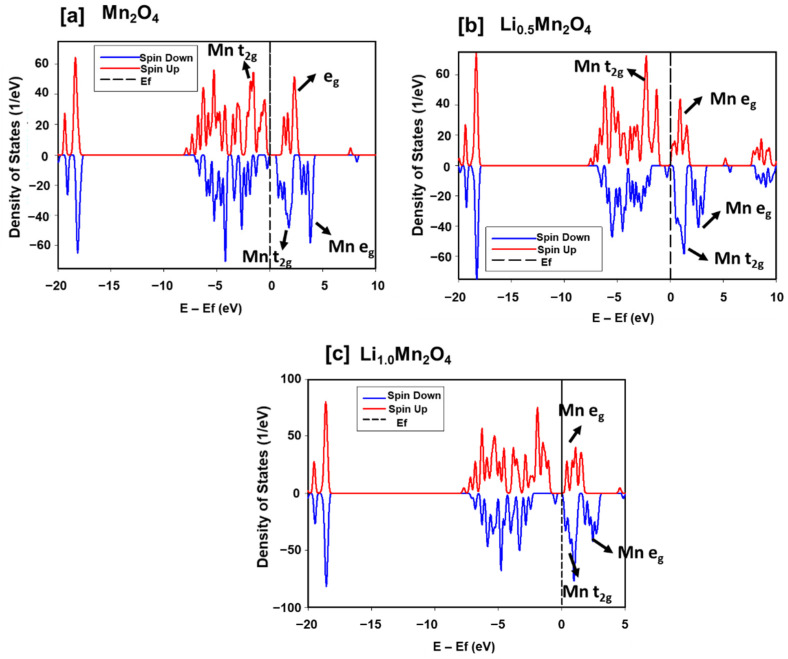
Spin up and spin down DoS of (**a**) Mn_2_O_4_, (**b**) Li_0.5_Mn_2_O_4_ and (**c**) Li_1.0_Mn_2_O_4_ calculated using ONETEP (Order-N Electronic Total Energy Package).

**Figure 9 materials-15-05678-f009:**
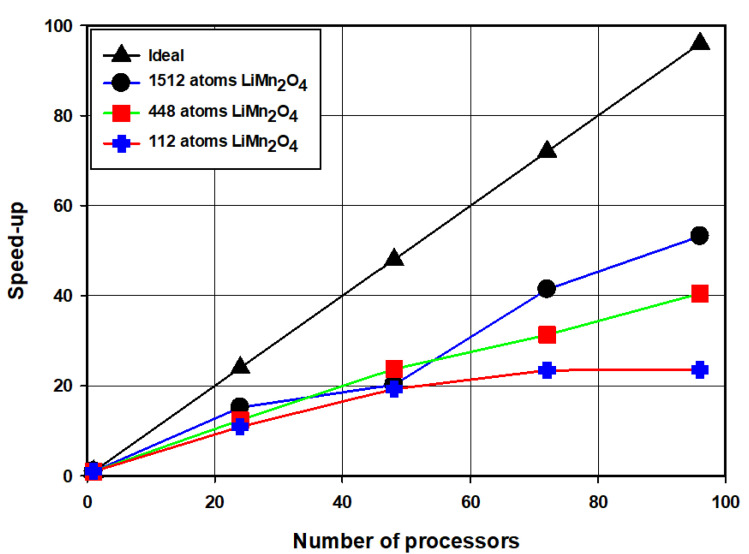
Strong scaling test on processors ranging from 1–96 for LiMn_2_O_4_ (112, 448, and 1512 atoms) compared to ideal linear scaling.

**Figure 10 materials-15-05678-f010:**
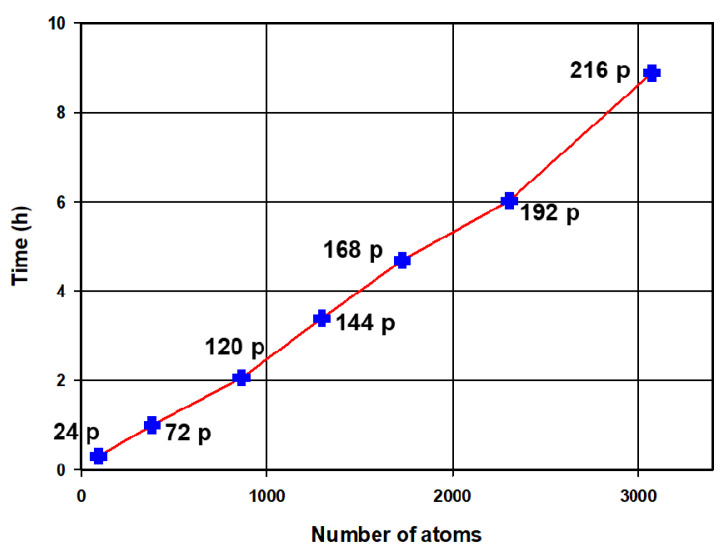
A plot of total computational time versus number of atoms for a LiMn_2_O_4_ spinel structure. The single point energy calculations were performed on varying computational resources. The number of processors were varied systematically with the number of atoms.

**Figure 11 materials-15-05678-f011:**
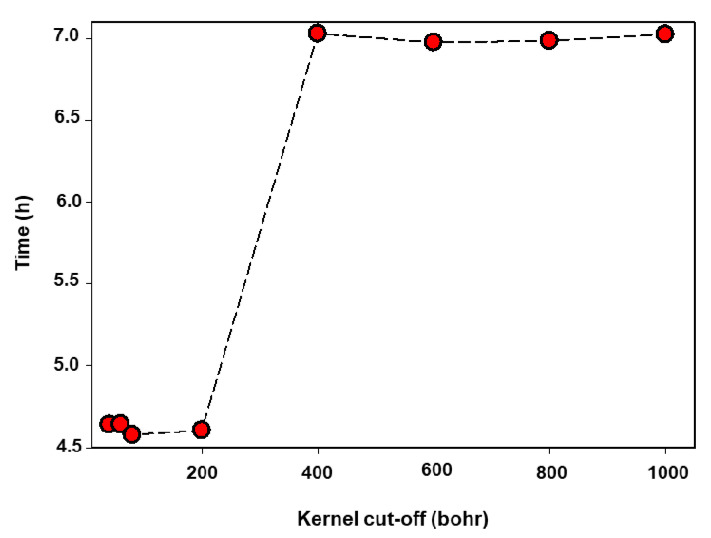
Investigation of the effect of kernel cut-off for a 1296 atoms Mn_2_O_4_ spinel structure on total computational time.

**Figure 12 materials-15-05678-f012:**
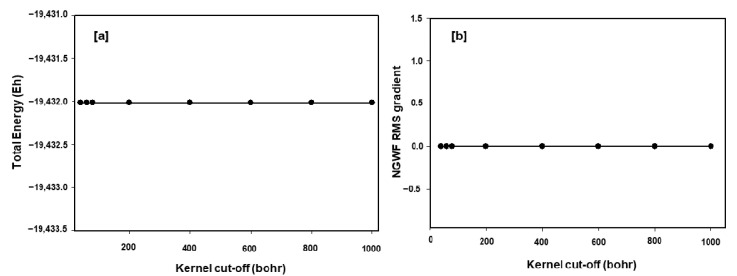
The figure shows the effect of the kernel cut-off on the (**a**) total energy and (**b**) NGWF RMS gradient for a 1296 atoms Mn_2_O_4_ spinel structure.

**Table 1 materials-15-05678-t001:** Lattice parameters of cubic spinel LiMn_2_O_4_.

Crystal Structure	Lattice Parameters (Å)
LiMn_2_O_4_ (This work—GGA)	8.297
LiMn_2_O_4_ ([30]—GGA)	8.160
LiMn_2_O_4_ ([31]—Experimental)	8.248
Li_2_Mn_2_O_4_ (This work—GGA)	8.231
Li_2_Mn_2_O_4_ ([32]—GGA)	8.380

## Data Availability

The data presented in this work are available on request from the corresponding author.

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
