# Peer review of "First-Principles Study on the Effect of Lithiation in Spinel LixMn2O4 (0 ≤ x ≤ 1) Structure: Calibration of CASTEP and ONETEP Simulation Codes"

_materials, 2022, doi:10.3390/ma15165678_

Round 1

Reviewer 1 Report

This manuscript reports some novel results of ONETEP Simulation Codes for the first-principles study of lithium-ion batteries. The results show that the ONETEP and CASTEP calculated results are in a great consonant, and through the scaling tests, it is proved that the use of ONETEP can greatly reduce the calculation time. The manuscript is well organized and completed, and the results are reasonable based on reliable analysis. However, some points should be addressed before being published. I believe this work is very convincing and impacting, but some minor changes should be addressed before being published.

  1. Was the same initial optimized LiMn2O4 configuration used in the comparative calculation of CASTEP and ONETEP? In other words, does the geometry optimization operate with CASTEP or ONETEP?
  2. How are the optimized Li1-xMn2O4 configurations obtained? The authors should give a description about the energy ranking and screening process.
  3. The DFT+U mode is crucial for the calculation of transition metals, does this work consider the Hubbard U correction?
  4. Some text in the figures is too small to read, the authors should make adjustments.

Reviewer 2 Report

Present work deals with computer modelling of Li-Mn-O system with the applying of CASTEP and ONETEP methods. The main structure and electronic properties were established. Discharging open-circuit voltages (OCVs) were also calculated. The following questions and comments appeared while reading the manuscript:

1. The topic of this article is very relevant, about a hundred publications on the topic of obtaining Li-Mn-O systems and their quantum calculations have been published in the last three years. It would be great if the authors expanded the introduction somewhat, taking into account current publications.

2. The authors calculated system LixMn2O4 with 0 ≤ x ≤ 1 and x=2. Why the analog systems with 1 < x < 2 were not taken in consideration?

3. The size of figure 2 should be increased.

4. In the section on structural parameters, the authors should specify the main parameters of the crystal lattice, as well as main geometric parameters such as the length of valence contacts and angles between atoms.

5. On the line 180 LixMn2O4 should be replaced by LixMn2O4.

6. In the conclusions section it is worth adding which of the simulated systems is the most promising for further study or potential application according to the author investigation.
